# Rapid encoding of task regularities in the human hippocampus guides sensorimotor timing

Ignacio Polti[1,2*†], Matthias Nau[1,2*†], Raphael Kaplan[1,3], Virginie van Wassenhove[4], Christian F Doeller[1,2,5*]

[1]Kavli Institute for Systems Neuroscience, Centre for Neural Computation, The Egil and Pauline Braathen and Fred Kavli Centre for Cortical Microcircuits, Jebsen Centre for Alzheimer's Disease, Norwegian University of Science and Technology, Trondheim, Norway; [2]Max Planck Institute for Human Cognitive and Brain Sciences, Leipzig, Germany; [3]Department of Basic Psychology, Clinical Psychology, and Psychobiology, Universitat Jaume I, Castellón de la Plana, Spain; [4]CEA DRF/Joliot, NeuroSpin; INSERM, Cognitive Neuroimaging Unit; CNRS, Université Paris-Saclay, Gif-Sur-Yvette, France; [5]Wilhelm Wundt Institute of Psychology, Leipzig University, Leipzig, Germany

**\*For correspondence:**
ignacio.polti@ntnu.no (IP);
matthias.nau@ntnu.no (MN);
doeller@cbs.mpg.de (CFD)

[†]These authors contributed equally to this work

**Abstract** The brain encodes the statistical regularities of the environment in a task-specific yet flexible and generalizable format. Here, we seek to understand this process by bridging two parallel lines of research, one centered on sensorimotor timing, and the other on cognitive mapping in the hippocampal system. By combining functional magnetic resonance imaging (fMRI) with a fast-paced time-to-contact (TTC) estimation task, we found that the hippocampus signaled behavioral feedback received in each trial as well as performance improvements across trials along with reward-processing regions. Critically, it signaled performance improvements independent from the tested intervals, and its activity accounted for the trial-wise regression-to-the-mean biases in TTC estimation. This is in line with the idea that the hippocampus supports the rapid encoding of temporal context even on short time scales in a behavior-dependent manner. Our results emphasize the central role of the hippocampus in statistical learning and position it at the core of a brain-wide network updating sensorimotor representations in real time for flexible behavior.

## Editor's evaluation

This important work brings ideas about hippocampal learning and involvement in temporal processing to a sensorimotor timing task, "time-to-contact estimation", that is not typically considered to be hippocampus-dependent. The study found that activity in the hippocampus measured with fMRI was related to feedback received about the accuracy of timing estimation and to performance improvement across trials in a manner not tied to the specific time interval tested. The evidence presented for the nature of the involvement of the hippocampus in this task is compelling.

## Introduction

When someone throws us a ball, we can anticipate its future trajectory, its speed and the time it will reach us. These expectations then inform the motor system to plan an appropriate action to catch it. Generating expectations and planning behavior accordingly builds on our ability to learn from past experiences and to encode the statistical regularities of the tasks we perform. At the core of this ability lies a continuous perception-action loop, initially proposed for sensorimotor systems (e.g.

*Wolpert et al., 2011*), which is now at the heart of many leading theories of brain function including active inference (*Friston et al., 2016*), predictive coding (*Huang and Rao, 2011*) and reinforcement learning (*Daw and Dayan, 2014*).

Critically, the brain needs to balance three primary objectives to effectively guide behavior in a dynamic environment. First, it needs to capture the specific aspects of the task that inform the relevant behavior (e.g. the remaining time to catch the ball). Second, it needs to generalize from a limited set of examples to novel and noisy situations. This can be achieved by regularizing the currently encoded information based on past experiences (e.g. by inferring how fast previous balls flew on average). Third, the sensorimotor representations that guide the behavior need to be updated flexibly whenever feedback about our actions becomes available (e.g. when we catch or miss the ball), or when task demands change (e.g. when someone throws us a frisbee instead). Herein, we refer to these objectives as specificity, regularization, and flexibility. While these are all fundamental principles underlying human cognition broadly, how the brain forms and continuously updates sensorimotor representations that balance these three objectives remains unclear.

Here, we approach this question with a new perspective by bridging two parallel lines of research centered on sensorimotor timing and hippocampal-dependent cognitive mapping. Specifically, we test how the human hippocampus, an area often implicated in episodic-memory formation (*Schiller et al., 2015*; *Eichenbaum, 2017*), may support the flexible updating of sensorimotor representations in real time and in concert with other regions. Importantly, the hippocampus is not traditionally thought to support sensorimotor functions, and its contributions to memory formation are typically discussed for longer time scales (hours, days, weeks). Here, however, we characterize in detail the relationship between hippocampal activity and real-time behavioral performance in a fast-paced timing task, which is traditionally believed to be hippocampal-independent. We propose that the capacity of the hippocampus to encode statistical regularities of our environment (*Doeller et al., 2005*; *Schapiro et al., 2017*; *Momennejad, 2020*) situates it at the core of a brain-wide network balancing specificity vs. regularization in real time as the relevant behavior is performed.

An optimal behavioral domain to study these processes is sensorimotor timing (*Gershman et al., 2014*; *Petter et al., 2018*). This is because prior work suggested that timing estimates indeed rely on the statistics of prior experiences (*Wolpert et al., 2011*; *Jazayeri and Shadlen, 2010*; *Acerbi et al., 2012*; *Chang and Jazayeri, 2018*). Crucially, however, timing estimates are not always accurate. Instead, they directly reflect the trade-off between specificity and regularization, which is expressed in systematic behavioral biases. Estimated intervals regress towards the mean of the distribution of tested intervals (*Jazayeri and Shadlen, 2010*), a well-known effect that we will refer to as the regression effect (*Petzschner et al., 2015*). The regression effect suggests that the brain encodes a probability distribution of possible intervals rather than the exact information obtained in each trial (*Wolpert et al., 2011*). Timing estimates therefore depend not only on the interval tested in a trial, but also on the temporal context in which they were encountered (i.e. the intervals tested in all other trials). This likely helps to predict future scenarios, to adapt behavior flexibly and to generalize to novel or noisy situations (*Jazayeri and Shadlen, 2010*; *Acerbi et al., 2012*; *Roach et al., 2017*).

Importantly, the hippocampus proper codes for time and temporal context on various scales (*Howard, 2017*) and it has been shown to process behavioral feedback in decision-making tasks (*Shohamy and Wagner, 2008*), pointing to a role in feedback learning. Moreover, the hippocampal formation has been implicated in encoding the latent structure of a task along with the individual features that were tested (*Kumaran, 2012*; *Schlichting and Preston, 2015*; *Schapiro et al., 2017*; *Wikenheiser et al., 2017*; *Behrens et al., 2018*; *Schuck and Niv, 2019*; *Whittington et al., 2020*; *Peer et al., 2021*), providing a unified account for its many proposed roles in navigation (*Burgess et al., 2002*), memory (*Schiller et al., 2015*; *Eichenbaum, 2017*) and decision making (*Kaplan et al., 2017*; *Vikbladh et al., 2019*). We propose that a central function of the human hippocampus is to encode the temporal context of stimuli and behaviors rapidly, and that this process manifests as the behavioral regression effect observed in time estimation and other domains (*Petzschner et al., 2015*). This puts the hippocampus at the core of a brain-wide network solving the trade-off between specificity and regularization for flexible behavior by continuously updating sensorimotor representations in a feedback-dependent manner. Using functional magnetic resonance imaging (fMRI) and a sensorimotor timing task, we here test this proposal empirically.

# Results

In the following, we present our experiment and results in four steps. First, we introduce our task, which built on the estimation of the time-to-contact (TTC) between a moving fixation target and a visual boundary, as well as the behavioral and fMRI measurements we acquired. On a behavioral level, we show that participants' timing estimates systematically regress towards the mean of the tested intervals. Second, we demonstrate that anterior hippocampal fMRI activity and functional connectivity tracks the behavioral feedback participants received in each trial, revealing a link between hippocampal processing and timing-task performance. Third, we show that this hippocampal feedback modulation reflects improvements in behavioral performance over trials. We interpret this activity to signal the updating of task-relevant sensorimotor representations in real time. Fourth, we show that these updating signals in the posterior hippocampus were independent of the specific interval that was tested and activity in the anterior hippocampus reflected the magnitude of the behavioral regression effect in each trial.

Notably, for each of the hippocampal main analyses, we also performed whole-brain voxel-wise analyses to uncover the larger brain network at play. We found that in addition to the hippocampus, regions typically associated with timing and reward processing signaled sensorimotor updating in our task, particularly the striatum. Follow-up analyses further revealed a striking distinction in TTC-specific and TTC-independent updating signals between striatal sub-regions. We conclude by discussing the potential neural underpinnings of these results and how the hippocampus may contribute to solving the trade-off between task specificity and regularization in concert with this larger brain network.

## Time-to-contact (TTC) estimation task

We monitored whole-brain activity using fMRI with concurrent eye tracking in 34 participants performing a TTC task. This task offered a rich behavioral read-out and required sustained attention in every single trial. During scanning, participants visually tracked a fixation target, which moved on linear trajectories within a circular boundary. The target moved at one of four possible speed levels and in one of 24 possible directions (*Figure 1A*, similar to *Nau et al., 2018a*). The sequence of tested speeds was counterbalanced across trials. Whenever the target stopped moving, participants estimated when the target would have hit the boundary if it had continued moving. They did so while maintaining fixation, and they indicated the estimated TTC by pressing a button. Feedback about their performance was provided foveally and instantly with a colored cue. The received feedback depended on the timing error, that is the difference between objectively true and estimated TTC (*Figure 1B*), and it comprised three levels reflecting high, middle, and low accuracy (*Figure 1C*). Because timing judgements typically follow the Weber-Fechner law (*Rakitin et al., 1998*), the feedback levels were scaled relative to the ground-truth TTC of each trial. This ensured that participants were exposed to approximately the same distribution of feedback at all intervals tested (*Figure 1C*, *Figure 1—figure supplement 1B*). After a jittered inter-trial interval (ITI), the next trial began and the target moved into another direction at a given speed. The tested speeds of the fixation target were counterbalanced across trials to ensure a balanced sampling within each scanning run. Because the target always stopped moving at the same distance to the boundary, matching the boundary's retinal eccentricity across trials, the different speeds led to four different TTCs: 0.55, 0.65, 0.86, and 1.2 s. Each participant performed a total of 768 trials. Please see Materials and methods for more details.

Analyzing the behavioral responses revealed that participants performed the task well and that the estimated and true TTCs were tightly correlated (*Figure 1B*; Spearman's $rho = 0.91, p = 2.2x10^{-16}$). However, participants' responses were also systematically biased towards the grand mean of the TTC distribution (0.82 s), indicating that shorter durations tended to be overestimated and longer durations tended to be underestimated. We confirmed this in all participants by examining the slopes of linear regression lines fit to the behavioral responses (*Figure 1—figure supplement 1D*). These slopes differed from 1 (veridical performance; *Figure 1B*, diagonal dashed line; one-tailed one-sample *t* test, $t(33) = -19.26, p = 2.2x10^{-16}, d = -3.30, CI : [-4.22, -2.47]$) as well as from 0 (grand mean; *Figure 1B*, horizontal dashed line; one-tailed one-sample *t* test, $t(33) = 21.62, p = 2.2x10^{-16}, d = 3.71, CI : [2.79, 4.72]$) and clustered at 0.5. Moreover, the slopes also correlated positively with behavioral accuracy across participants (*Figure 1—figure supplement 1E*; Spearman's $rho = 0.794, p = 2.1x10^{-08}$), consistent with previous reports (*Cicchini et al., 2012*). Notably, the regression effect we observed in behavior has been argued to show that timing estimates indeed rely on the latent task regularities that our brain has

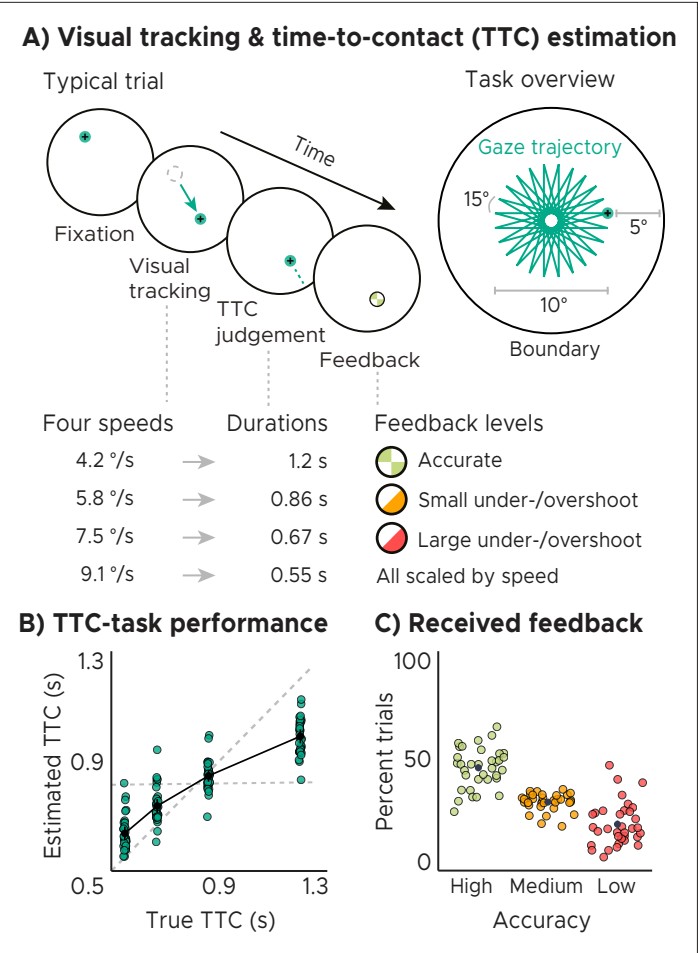

**Figure 1.** Visual tracking and Time-To-Contact (TTC) estimation task. (**A**) Task design. In each trial during fMRI scanning, participants fixated a target (phase 1), which started moving at one of 4 possible speeds and in one of 24 possible directions for 10° visual angle (phase 2). After the target stopped moving, participants kept fixating and estimated when the fixation target would have hit a boundary 5° visual angle apart (phase 3). After pressing a button at the estimated TTC, participants received feedback (phase 4) according to their performance. Feedback was scaled relative to target TTC. (**B**) Task performance. True and estimated TTC were correlated, showing that participants performed the task well. However, they overestimated short TTCs and underestimated long TTCs. Their estimates regressed towards the grand-mean of the TTC distribution (horizontal dashed line), away from the line of equality (diagonal dashed line). (**C**) Feedback. On average, participants received high-accuracy feedback on half of the trials (also see *Figure 1—figure supplement 1B*, *Figure 1—figure supplement 1C*). (BC) We plot the mean and SEM (black dots and lines) as well as single-participant data as dots (n=34). Feedback levels are color coded.

The online version of this article includes the following figure supplement(s) for figure 1:

**Figure supplement 1.** Behavioral analyses.

**Figure supplement 2.** Eye tracking analyses.

encoded (e.g. *Jazayeri and Shadlen, 2010*; *Roach et al., 2017*). It may therefore reflect a key behavioral adaptation helping to regularize encoded intervals to optimally support both current task performance and generalization to future scenarios. In support of this, participants' regression slopes converged over time towards the optimal value of 0.5, that is the slope value between veridical performance and the grand mean (*Figure 1—figure supplement 1F*; linear mixed-effects model with task segment as a predictor and participants as the error term, $F(1) = 8.172, p = 0.005, \epsilon^2 = 0.08, CI : [0.01, 0.18]$), and participants' slope values became more similar (*Figure 1—figure supplement 1G*; linear regression with task segment as predictor, $F(1) = 6.283, p = 0.046, \epsilon^2 = 0.43, CI : [0, 1]$). Consequently, this also led to an improvement in task performance over time on group level (i.e. task accuracy and

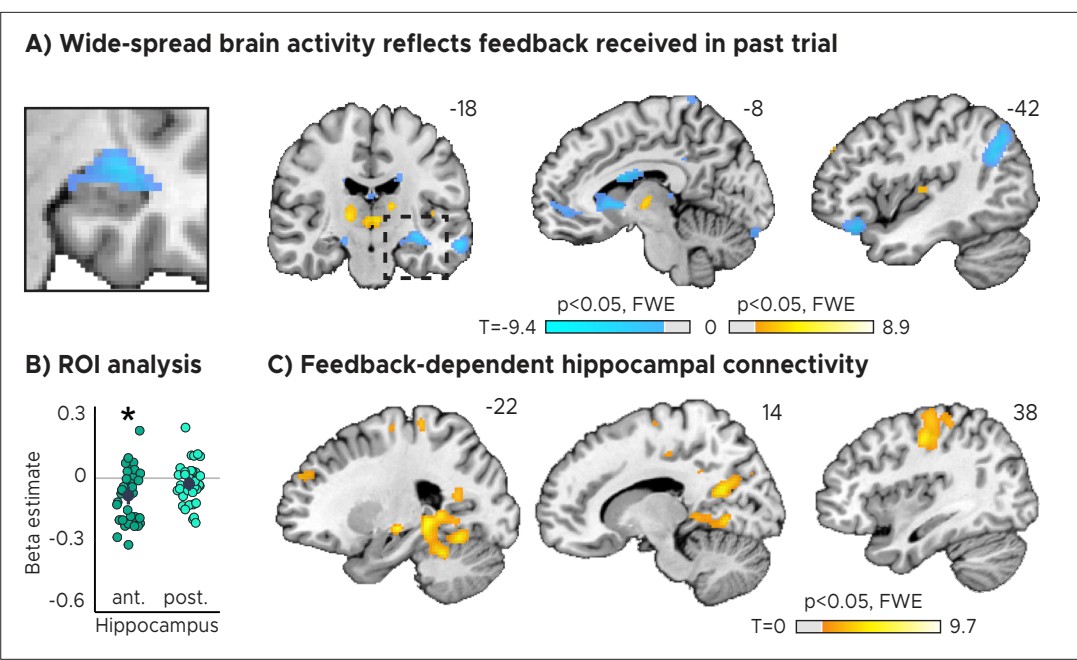

**Figure 2.** Feedback on the previous trial (n-1) modulates network-wide activity and hippocampal connectivity in subsequent trials (n). (**A**) Voxel-wise analysis. Activity in each trial was modeled with a separate regressor as a function of feedback received in the previous trial. Insert zooming in on hippocampus added. (**B**) Independent regions-of-interest analysis for the anterior (ant.) and posterior (post.) hippocampus. We plot the beta estimates obtained for the contrast between low-accuracy vs. high-accuracy feedback. Negative values indicate that smaller errors, and higher-accuracy feedback, led to stronger activity. Depicted are the mean and SEM across participants (black dot and line) overlaid on single participant data (coloured dots; n=34). Activity in the anterior hippocampus is modulated by feedback received in previous trial. Statistics reflect p<0.05 at Bonferroni-corrected levels (*) obtained using a group-level two-tailed one-sample t-test against zero. (**C**) Feedback-dependent hippocampal connectivity. We plot results of a psychophysiological interactions (PPI) analysis conducted using the hippocampal peak effects in (**A**) as a seed for low vs. high-accuracy feedback. (AC) We plot thresholded t-test results at 1 mm resolution overlaid on a structural template brain. MNI coordinates added. Hippocampal activity and connectivity is modulated by feedback received in the previous trial.

The online version of this article includes the following figure supplement(s) for figure 2:

**Figure supplement 1.** Regions of interest (ROIs).

**Figure supplement 2.** Current trial effects.

**Figure supplement 3.** Brain activity reflects feedback received in past trial.

**Figure supplement 4.** Remaining contrasts from *Figure 2A*, *Figure 2B*.

precision increased; *Figure 1—figure supplement 1I*), and the relationship between accuracy and precision became stronger (*Figure 1—figure supplement 1H*), linear mixed-effect model results for accuracy: $F(1) = 15.127, p = 1.3x10^{-4}, \epsilon^2 = 0.06, CI : [0.02, 0.11]$, precision: $F(1) = 20.189, p = 6.1x10^{-5}, \epsilon^2 = 0.32, CI : [0.13, 1]$, accuracy-precision relationship: $F(1) = 8.288, p = 0.036, \epsilon^2 = 0.56, CI : [0, 1]$, see methods for model details.

## Behavioral feedback predicts hippocampal activity

Importantly, sensorimotor updating is expected to occur right after the value of the performed action became apparent, which is when participants received feedback. As a proxy, we therefore analyzed how activity in each voxel reflected the feedback participants received in the previous trial. Using a mass-univariate general linear model (GLM), we modeled the three feedback levels with one regressor each plus additional nuisance regressors (see Materials and methods for details). The three feedback levels (high, medium, and low accuracy) corresponded to small, medium and large timing errors, respectively. We then contrasted the beta weights estimated for low-accuracy vs. high-accuracy feedback and examined the effects on group-level averaged across runs. We performed both whole-brain

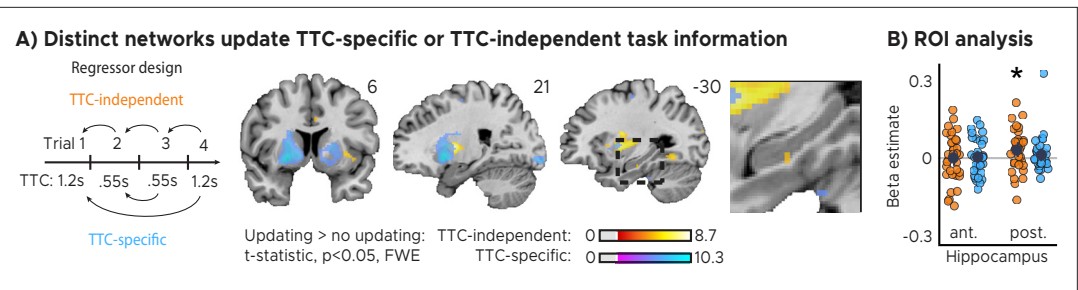

**Figure 3.** Distinct cortical and subcortical networks signal the updating of TTC-specific and TTC-independent task information. (**A**) Left panel: Visual depiction of parametric modulator design. Two regressors per run modeled the improvement in behavioral performance since the last trial independent of the tested TTC (Regressor 1: TTC-independent) or the improvement since the last trial when the same target TTC was tested (Regressor 2: TTC-specific). Right panel: Voxel-wise analysis results for TTC-specific and TTC-independent regressors. We plot thresholded t-test results at 1 mm resolution at p<0.05 whole-brain Family-wise-error (FWE) corrected levels overlaid on a structural template brain. Insert zooming in on hippocampus and MNI coordinates added. (**B**) Independent regions-of-interest analysis for the anterior (ant.) and posterior (post.) hippocampus. We plot the beta estimates obtained for TTC-independent in orange and TTC-specific regressors in blue. Depicted are the mean and SEM across participants (black dot and line) overlaid on single participant data as dots (n=34). Statistics reflect p<0.05 at Bonferroni-corrected levels (*) obtained using a group-level one-tailed one-sample t-test against zero.

The online version of this article includes the following figure supplement(s) for figure 3:

**Figure supplement 1.** Distinct networks support TTC-specific and TTC-independent updating.

**Figure supplement 2.** TTC-independent hippocampal connectivity.

voxel-wise analyses as well as regions-of-interest (ROI) analysis for anterior and posterior hippocampus separately, for which prior work suggested functional differences with respect to their contributions to memory-guided behavior (*Poppenk et al., 2013*; *Strange et al., 2014*).

In both our ROI analysis and a voxel-wise analysis, we found that hippocampal activity could indeed be predicted by the feedback participants received in the previous trial (*Figure 2A*, *Figure 2B*). Higher-accuracy feedback resulted in overall stronger activity in the anterior section of the hippocampus (*Figure 2B*, *Figure 2—figure supplement 1A*; two-tailed one-sample $t$ tests: anterior HPC, $t(33) = -3.80, p = 5.9x10^{-4}, p_{fwe} = 0.001, d = -0.65, CI : [-1.03, -0.28]$; posterior HPC, $t(33) = -1.60, p = 0.119, p_{fwe} = 0.237, d = -0.27, CI : [-0.62, 0.07]$). Moreover, the voxel-wise analysis revealed feedback-related activity also in the thalamus and the striatum (*Figure 2A*), and in the hippocampus when the feedback of the current trial was modeled (*Figure 2—figure supplement 2A*).

Note that these results were robust even when fewer nuisance regressors were included to control for model over-specification (*Figure 2—figure supplement 3B*; two-tailed one-sample $t$ tests: anterior HPC, $t(33) = -3.65, p = 8.9x10^{-4}, p_{fwe} = 0.002, d = -0.63, CI : [-1.01, -0.26]$; posterior HPC, $t(33) = -1.43, p = 0.161, p_{fwe} = 0.322, d = -0.25, CI : [-0.59, 0.10]$), and when all three feedback levels were modeled with one parametric regressors (*Figure 2—figure supplement 3C*; two-tailed one-sample $t$ tests: anterior HPC, $t(33) = -3.59, p = 0.002, p_{fwe} = 0.005, d = -0.56, CI : [-0.93, -0.20]$; posterior HPC, $t(33) = -0.99, p = 0.329, p_{fwe} = 0.659, d = -0.17, CI : [-0.51, 0.17]$). In addition, hippocampal activity was higher for medium-accuracy feedback relative to low-accuracy feedback on voxel-wise and ROI level (*Figure 2—figure supplement 4A*; two-tailed one-sample $t$ tests: anterior HPC, $t(33) = -4.40, p = 1.110^{-4}, p_{fwe} = 2.110^{-4}, d = -0.76, CI : [-1.15, -0.37]$; posterior HPC, $t(33) = -3.62, p = 9.810^{-4}, p_{fwe} = 0.002, d = -0.62, CI : [-1.00, -0.25]$) and for high-accuracy feedback vs. medium-accuracy feedback on voxel-wise but not ROI level (*Figure 2—figure supplement 4B*; two-tailed one-sample $t$ tests: anterior HPC, $t(33) = -0.08, p = 0.933, p_{fwe} = 1, d = -0.01, CI : [-0.36, 0.33]$; posterior HPC, $t(33) = t = 0.99, p = 0.327, p_{fwe} = 0.654, d = 0.17, CI : [-0.17, 0.52]$). Further, there was no systematic relationship between subsequent trials on a behavioral level (*Figure 1—figure supplement 1A*; two-tailed one-sample $t$ test; $t(33) = 1.03, p = 0.312, d = 0.18, CI : [-0.17, 0.52]$; see Materials and methods for details) and that the direction of the effects differed across regions (*Figure 2A*), speaking against potential feedback-dependent biases in attention.

## Feedback-dependent hippocampal functional connectivity

Having established that hippocampal activity reflected feedback in the TTC task, we reasoned that its activity may also show systematic co-fluctuations with other task-relevant brain regions as well. To test this, we estimated the functional connectivity of a 4 mm radius sphere centered on the hippocampal peak main effect (x=-32, y=-14, z=-14) using a seed-based psychophysiological interaction (PPI) analysis (see Materials and methods). We reasoned that larger timing errors and therefore low-accuracy feedback would result in stronger updating compared to smaller timing errors and high-accuracy feedback, a relationship that should also be reflected in the functional connectivity between the hippocampus and other regions. We specifically tested this using the PPI analysis by contrasting trials in which participants performed poorly compared to those trials in which they performed well.

We found that hippocampal activity co-fluctuated with activity in the primary motor cortex, the parahippocampal gyrus and medial parietal lobe as well as the cerebellum (*Figure 2C*). These co-fluctuations were stronger when participants performed poorly in the previous trial and therefore when they received low-accuracy feedback. Combined with the previous analysis, this means that the absolute hippocampal activity scaled positively (*Figure 2A*, *Figure 2B*) and functional connectivity scaled negatively (*Figure 2C*) with feedback valence.

## Hippocampal activity explains accuracy and biases in task performance

Two critical open questions remained. First, did the observed feedback modulation actually reflect improvements in behavioral performance over trials? Second, was the information that was learned specific to the interval that was tested in a given trial, likely serving task specificity, or was independent of the tested interval, potentially serving regularization? To answer these questions in one analysis, we used a GLM modeling activity not as a function of feedback received in the previous trial (*Figure 2*), but as a function of the difference in feedback between trials (*Figure 3*). Specifically, we modeled with two separate parametric regressors the improvements in TTC task performance across subsequent trials (regressor 1: TTC-independent updating) as well as the improvements over subsequent trials in which the same TTC interval was tested (regressor 2: TTC-specific updating). We again accounted for nuisance variance as before, and we contrasted trials in which participants had improved versus the ones in which they had not improved or got worse (see Materials and methods for details). Because stronger sensorimotor updating should lead to larger performance improvements, we predicted to find stronger activity for improvements vs. no improvements in these tests (one-tailed hypothesis).

We found both TTC-specific and TTC-independent activity throughout cortical and subcortical regions. Distinct areas engaged in either one or in both of these processes (*Figure 3A*, *Figure 3—figure supplement 1*). Crucially, we found that hippocampal activity signaled behavioral improvements independent of the TTC intervals tested. This effect was localized to the posterior section of the hippocampus (*Figure 3B*, *Figure 2—figure supplement 1A*; one-tailed one-sample $t$ tests; TTC-independent: anterior HPC, $t(33) = 0.36, p = 0.360$, $p_{fwe} = 1, d = 0.06, CI : [-0.28, 0.40]$, posterior HPC, $t(33) = 2.81, p = 0.004$, $p_{fwe} = 0.017, d = 0.48, CI : [0.12, 0.85]$; TTC-specific: anterior HPC, $t(33) = 0.57, p = 0.285$, $p_{fwe} = 1, d = 0.10, CI : [-0.24, 0.44]$, posterior HPC, $t(33) = 1.29, p = 0.103$, $p_{fwe} = 0.413, d = 0.22, CI : [-0.12, 0.57]$). We then again estimated the functional connectivity profile of the hippocampal main effect using a PPI analysis (sphere with 4 mm radius centered on the peak voxel at x=-30, y=-24, z=-18), revealing co-fluctuations in multiple regions including the putamen and the thalamus that were specific to behavioral improvements (*Figure 3—figure supplement 2*).

We reasoned that updating TTC-independent information may support generalization performance by means of regularizing the encoded intervals based on the temporal context in which they were encoded. In our task, an efficient way of regularizing the encoded information is to bias one's TTC estimates towards the mean of the TTC distribution, which corresponds to the regression effect that we observed on a behavioral level (*Figure 1B*, *Figure 1—figure supplement 1D*). Given the hippocampal feedback modulation and updating activity we reported above, we hypothesized that hippocampal activity should therefore also reflect the magnitude of the regression effect in behavior. To test this in a final analysis, we modeled the activity in each trial parametrically either as a function of performance (i.e. the absolute difference between estimated and true TTC) or as a function of the strength of the regression effect in each trial (i.e. the absolute difference between the estimated TTC and the mean of the tested intervals). Voxel-wise weights for these two regressors were estimated in two independent GLMs (see Materials and methods for details).

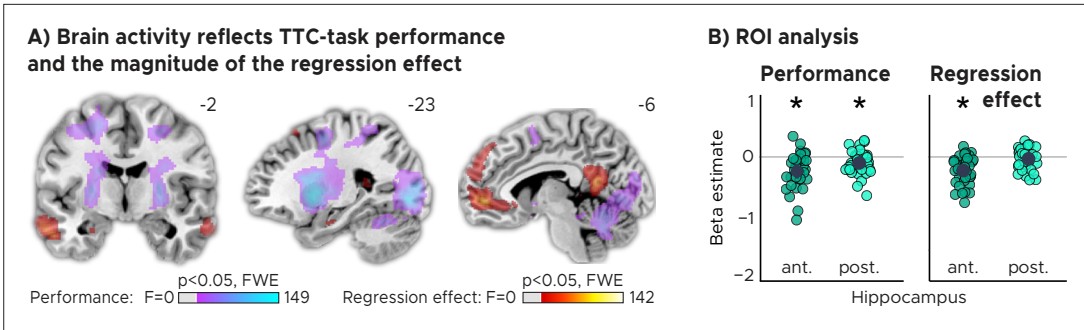

**Figure 4.** TTC-task performance vs. behavioral regression effect. (**A**) Voxel-wise analysis. We plot thresholded F-test results for the task-performance regressor and the regression-to-the-mean regressor at 1 mm resolution overlaid on a structural template brain. MNI coordinates added. Distinct networks reflect task performance and the magnitude of the regression effect. (**B**) Independent regions-of-interest analysis for the anterior (ant.) and posterior (post.) hippocampus. We plot the beta estimates obtained for each participant for each of the two regressors. Negative values indicate a linear increase between hippocampal activity and either task performance (left, Performance) or the magnitude of the regression effect (right, Regression effect). Depicted are the mean and SEM across participants (black dot and line) overlaid on single participant data (colored dots; n=34). Statistics reflect p<0.05 at Bonferroni-corrected levels (*) obtained using a group-level two-tailed one-sample t-test against zero.

Our analyses showed that trial-wise hippocampal activity increased with better TTC-task performance (*Figure 4A*, *Figure 4B*; two-tailed one-sample $t$ tests; anterior HPC, $t(33) = -4.85, p = 2.9x10^{-5}$, $p_{fwe} = 5.8x10^{-5}$, $d = -0.83, CI: [-1.24, -0.44]$; posterior HPC, $t(33) = -2.88, p = 0.007$, $p_{fwe} = 0.014, d = -0.49$, $CI: [-0.86, -0.14]$), and consistently also with the valence of the feedback received in the current trial (*Figure 2—figure supplement 2*). In addition, however, and as predicted, it also reflected the trial-wise magnitude of the behavioral regression effect (*Figure 4A*, *Figure 4B*; two-tailed one-sample $t$ tests; anterior HPC, $t(33) = -5.55, p = 3.6x10^{-6}, p_{fwe} = 1.1x10^{-5}, d = -0.95, CI: [-1.37, -0.55]$; posterior HPC, $t(33) = -1.06, p = 0.295, p_{fwe} = 0.886, d = -0.18, CI: [-0.53, 0.16]$). Activity in the anterior hippocampus was stronger in trials in which participants' TTC estimates were more biased towards the mean of the sampled intervals (indicated by a negative beta estimate). Notably, similar effects were observed in prefrontal and posterior cingulate areas (*Figure 4A*).

### Eye tracking: no relevant biases in viewing behavior

To ensure that our results could not be attributed to systematic error patterns in viewing behavior, we analyzed the co-recorded eye tracking data of our participants in detail. After data cleaning (see Materials and methods), we used Wilcoxon signed-rank tests for paired samples to control for differences in fixation accuracy across speed levels (*Figure 1—figure supplement 2A*; 4.17°/s vs.5.81°/s: $V = 171, p = 0.03, p_{fwe} = 0.179, r = -0.04, CI: [-0.31, 0.23]$; 4.17°/s vs.7.45°/s: $V = 152, p = 0.012, p_{fwe} = 0.071, r = -0.06, CI: [-0.32, 0.22]$; 4.17°/s vs.9.09°/s: $V = 161, p = 0.019, p_{fwe} = 0.112, r = -0.08, CI: [-0.34, 0.2]$; 5.81°/s vs.7.45°/s: $V = 224, p = 0.215, p_{fwe} = 1, r = -0.01, CI: [-0.28, 0.26]$; 5.81°/s vs.9.09°/s: $V = 217, p = 0.174, p_{fwe} = 1, r = -0.04, CI: [-0.31, 0.23]$; 7.45°/s vs.9.09°/s: $V = 263, p_{fwe} = 1$ $p = 0.566., r = -0.01, CI: [-0.28, 0.26]$) and accuracy levels (*Figure 1—figure supplement 2B*; Low *vs.* Medium: $V = 380, p = 0.163, p_{fwe} = 0.489, r = 0.02, CI: [-0.25, 0.29]$; Low *vs.* High: $V = 366, p = 0.249, p_{fwe} = 0.747, r = 0.03, CI: [-0.24, 0.3]$; Medium *vs.* High: $V = 278, p = 0.748, p_{fwe} = 1, r = -0.04, CI: [-0.31, 0.23]$). Moreover, we examined the relationship of the fixation error with TTC-task performance (*Figure 1—figure supplement 2C*; Spearman's $rho = 0.17, p = 0.344$) as well as with the behavioral regression effect (*Figure 1—figure supplement 2C*; Spearman's $rho = 0.26, p = 0.131$). None of these control analyses suggested that biased patterns in viewing behavior could hinder the interpretation of our results.

### Discussion

This study investigated how the human brain flexibly updates sensorimotor representations in a feedback-dependent manner in the service of timing behavior. We specifically focused on the hippocampus, due to its known role in temporal coding and learning, asking how hippocampal processing

may support behavioral flexibility, specificity, and regularization. Because anterior and posterior sections of the hippocampus differ in whole-brain connectivity as well as in their contributions to memory-guided behavior (*Strange et al., 2014*), we analyzed the two sections separately. Moreover, we explored the larger brain-wide network involved in balancing these objectives. To do so, we monitored human brain activity with fMRI while participants estimated the time-to-contact between a moving target and a visual boundary. This allowed us to analyze brain activity as a function of task performance and as a function of the improvements in performance over time. We found that anterior hippocampal activity as well as functional connectivity reflected the feedback participants received during this task, and its activity followed the performance improvements in a temporal-context-dependent manner. Its activity reflected trial-wise behavioral biases towards the mean of the sampled intervals, and activity in the posterior hippocampus signaled sensorimotor updating independent of the specific intervals tested. In what follows, we discuss our results in the context of prior work on timing behavior and on hippocampal spatiotemporal coding. Moreover, we elaborate on the domain-general nature of hippocampal-cortical interactions and on the sensorimotor updating mechanisms that potentially underlie the effects observed in this study.

## Spatiotemporal coding in the hippocampus

The hippocampus encompasses neurons sensitive to elapsed time (*Paton and Buonomano, 2018*; *Eichenbaum, 2014*; *Umbach et al., 2020*). These cells might play an important role in guiding timing behavior (*Nobre and van Ede, 2018*), which potentially explains why hippocampal damage or inactivation impairs the ability to estimate durations in rodents (*Meck et al., 1984*) and humans (*Richards, 1973*). Our results are in line with these reports, showing that hippocampal fMRI activity also reflects participants' TTC estimation ability (*Figure 4*). They are also in line with other human neuroimaging studies suggesting that the hippocampus bridges temporal gaps between two stimuli during trace eyeblink conditioning (*Cheng et al., 2008*), and that it represents duration within event sequences (*Barnett et al., 2014*; *Thavabalasingam et al., 2018*; *Thavabalasingam et al., 2019*).

Our results speak to the above-mentioned reports by revealing that the hippocampus is an integral part of a widespread brain network contributing to sensorimotor updating of encoded intervals in humans (*Figure 2*, *Figure 3*, *Figure 4*, *Figure 2—figure supplement 2*, *Figure 3—figure supplement 1*, *Figure 3—figure supplement 2*). Moreover, they demonstrate a direct link between hippocampal activity, the feedback participants received and the behavioral improvements expressed over time (*Figure 3*), emphasizing its role in feedback learning. Critically, the underlying process must occur in real-time when feedback is presented, suggesting that it plays out on short time scales. Notably, the human hippocampus is neither typically linked to sensorimotor timing tasks such as ours, nor is its activity considered to reflect temporal relationships on such short time scales. Instead, human hippocampal processing is often studied in the context of much longer time scales (*Schiller et al., 2015*; *Eichenbaum, 2017*), which showed that it may support the encoding of the progression of events into long-term episodic memories (*Deuker et al., 2016*; *Montchal et al., 2019*; *Bellmund et al., 2022*) or contribute to the establishment of chronological relations between events in memory (*Gauthier et al., 2019*; *Gauthier et al., 2020*). Intriguingly, the mechanisms at play may build on similar temporal coding principles as those discussed for motor timing (*Yin and Troger, 2011*; *Eichenbaum, 2014*; *Howard, 2017*; *Palombo and Verfaellie, 2017*; *Nobre and van Ede, 2018*; *Paton and Buonomano, 2018*; *Bellmund et al., 2020*; *Bellmund et al., 2022*; *Shikano et al., 2021*; *Shimbo et al., 2021*), with differential contributions of the anterior and posterior hippocampus. Note that our observation of distinct activity modulations in the anterior and posterior hippocampus suggests that the functions and coding principles discussed here may be mediated by at least partially distinct populations of hippocampal cells.

Our task can be solved by estimating temporal intervals directly, but also by extrapolating the movement of the fixation target over time, shifting the locus of attention towards the target boundary (*Figure 1*). The brain may therefore likely monitor the temporal and spatial task regularities in parallel. Participants' TTC estimates were further informed exclusively by the speed of the target, which inherently builds on tracking kinematic information over time, which may explain why TTC tasks also engage visual motion regions in humans (*de Azevedo Neto and Amaro Júnior, 2018*). While future studies could tease apart spatial and temporal factors explicitly, our results are in line with both accounts. For example, the hippocampus and surrounding structures represent maps of visual space in primates,

which potentially mediate a coordinate system for planning behavior, integrating visual information with existing knowledge and to compute vectors in space (*Nau et al., 2018b*; *Bicanski and Burgess, 2020*). These visuospatial representations are perfectly suited to guide attention and therefore the relevant behaviors in our task (*Aly and Turk-Browne, 2017*), which could be tested in the future akin to prior work using a similar paradigm (*Nau et al., 2018a*).

## The role of feedback in timed motor actions

Importantly, our results neither imply that the hippocampus acts as an 'internal clock', nor do we think of it as representing action sequences or coordinating motor commands directly. Rather, its activity may indicate the feedback-dependent updating of encoded information more generally and independent of the task that was used. The hippocampal formation has been proposed as a domain-general learning system (*Kumaran, 2012*; *Schlichting and Preston, 2015*; *Chersi and Burgess, 2015*; *Schapiro et al., 2017*; *Wikenheiser et al., 2017*; *Behrens et al., 2018*; *Bellmund et al., 2018*; *Vikbladh et al., 2019*; *Geerts et al., 2020*; *Momennejad, 2020*; *Bellmund et al., 2022*), which may encode the structure of a task abstracted away from our immediate experience. In contrast, the striatum was proposed to encode sensory states or actions, supporting the learning of task-specific (egocentric) information (*Chersi and Burgess, 2015*; *Geerts et al., 2020*). Together, the two regions may therefore play an important role in decision making in general also in other non-temporal domains.

Consistent with these ideas, we observed that striatal and hippocampal activity was modulated by behavioral feedback received in each trial (*Figure 2*, *Figure 2—figure supplement 1*). Similar feedback signals have been previously linked to learning (*Schönberg et al., 2007*; *Cohen and Ranganath, 2007*; *Shohamy and Wagner, 2008*; *Foerde and Shohamy, 2011*; *Wimmer et al., 2012*) and the successful formation of hippocampal-dependent long-term memories in humans (*Wittmann et al., 2005*). Moreover, hippocampal activity is known to signal learning in other tasks (*Doeller et al., 2008*; *Foerde and Shohamy, 2011*; *Dickerson and Delgado, 2015*; *Wirth et al., 2009*; *Schapiro et al., 2017*; *Kragel et al., 2021*). Here, we show a direct relationship between hippocampal activity and ongoing timing behavior, and we show that receiving behavioral feedback modulates widespread brain activity (*Figure 2*, *Figure 2—figure supplement 1*), which potentially reflects the involvement of these areas in the coordination of reward behavior observed earlier (*LeGates et al., 2018*). These regions include those serving sensorimotor functions, but also those encoding the structure of a task or the necessary value functions associated with specific actions (*Lee et al., 2012*).

The present study further demonstrates that activity in the hippocampus co-fluctuates with activity in other likely task-relevant regions in a task-dependent manner. We observed such co-fluctuations in the striatum and cerebellum, often associated with reward processing and action coordination (*Bostan and Strick, 2018*; *Cox and Witten, 2019*), the motor cortex, typically involved in action planning and execution, as well as the parahippocampal gyrus and medial parietal lobe, often associated with visual-scene analysis (*Epstein and Baker, 2019*). This may indicate that behavioral feedback also affects the functional connectivity profile of the hippocampus with those domain-selective regions that are currently engaged in the ongoing task. In the present report, this included the motor cortex, the parahippocampal gyrus, the medial parietal lobe and the cerebellum. This may also relate to previous reports of the cerebellum contributing temporal signals to cortical regions during similar tasks as ours (*O'Reilly et al., 2008*). Interestingly, we observed that functional connectivity of the anterior hippocampus scaled negatively (*Figure 2C*) with feedback valence, unlike its absolute activity, which scaled positively with feedback valence (*Figure 2A*, *Figure 2B*), suggesting that the two measures may be sensitive to related but distinct processes.

What might be the neural mechanism underlying sensorimotor updating signals in our task? Prior work has shown that hippocampal, frontal and striatal temporal receptive fields scale relative to the tested intervals, and that they re-scale dynamically when those tested intervals change (*MacDonald et al., 2011*; *Gouvêa et al., 2015*; *Mello et al., 2015*; *Wang et al., 2018*). This may enable the encoding and continuous maintenance of optimal task priors, which keep our actions well-adjusted to our current needs. We speculate that such receptive-field re-scaling also underlies the continuous updating effects discussed here. Consistent with this idea and the present results, receptive-field re-scaling can occur on a trial-by-trial basis in the hippocampus (*Shikano et al., 2021*; *Shimbo et al., 2021*) but also in other regions such as the striatum and frontal cortex (*Mello et al., 2015*; *Gouvêa et al., 2015*; *Wang et al., 2018*). Such network-wide receptive-field re-scaling likely builds on a re-weighting

of functional connections between neurons and regions, which may explain why anterior hippocampal connectivity correlated negatively with feedback valence in our data. Larger errors may have led to stronger re-scaling, which may be grounded in a corresponding change in functional connectivity.

## A trade-off between specificity and regularization?

So far, we discussed how the brain may capture the temporal structure of a task and how the hippocampus supports this process. However, how do we encode specific task details while still forming representations that generalize well to new scenarios? In other words, how does the brain encode the probability distribution of the intervals we tested optimally without overfitting? Our behavioral and neuroimaging results suggest that this trade-off between specificity and regularization is governed by many regions, updating different types of task information in parallel (*Figure 3A*). For example, hippocampal activity reflected performance improvements independent of the tested interval, whereas the caudate signaled improvements specifically over those trials in which the same TTC was tested. In the putamen, we found evidence for both processes (*Figure 3—figure supplement 1B*). This suggests that different regions encode distinct task regularities in parallel to form optimal sensorimotor representations to balance specificity and regularization. This is in line with our behavioral results, showing that TTC-task performance became more optimal in the face of both of these two objectives. Over time, behavioral responses clustered more closely between the diagonal and the average line in the behavioral response profile (*Figure 1B*, *Figure 1—figure supplement 1G*), and the TTC error decreased over time. While different participants approached these optimal performance levels from different directions, either starting with good performance or strong regularization, the group approached overall optimal performance levels over the course of the experiment.

Because hippocampal activity (*Julian and Doeller, 2020*) and the regression effect (*Jazayeri and Shadlen, 2010*) were previously linked to the encoding of context, we reasoned that hippocampal activity should also be related to the regression effect directly. This may explain why hippocampal activity reflected the magnitude of the regression effect as well as behavioral improvements independently from TTC, and why it reflected feedback, which informed the updating of the internal prior. Notably, our results make a central prediction for future research. We anticipate that participants with stronger updating activity in the hippocampus should be able to generalize better to new scenarios, for example when new intervals are tested. While we could not test this prediction directly in our study, we did test for a link to a related phenomenon, and that is the regression effect we observed on the behavioral level. We found that TTC estimates regressed towards the mean of the sampled intervals in all participants (*Figure 1B*, *Figure 1—figure supplement 1D*), an effect that is well known in the timing literature (*Jazayeri and Shadlen, 2010*) and other domains (*Petzschner and Glasauer, 2011*; *Petzschner et al., 2015*). This regression effect likely reflects regularization in support of generalization (*Roach et al., 2017*), because time estimates are biased towards the mean of the tested intervals, and because the mean will likely be close to the mean of possible future intervals. We therefore hypothesized that this effect is grounded in the activity of the hippocampus, because it plays a central role in generalization in other non-temporal domains (*Kumaran, 2012*; *Schlichting and Preston, 2015*; *Schapiro et al., 2017*; *Momennejad, 2020*). Our analyses revealed that this was indeed the case. We found that hippocampal activity followed the magnitude of the regression effect in each trial (*Figure 4*), potentially reflecting the temporal-context-dependent regularization of encoded intervals toward the grand mean of the tested intervals (*Jazayeri and Shadlen, 2010*).

In addition, our voxel-wise results showed that striatal subregions only tracked how accurate participants' responses were, not how strongly they regressed towards the mean (*Figure 4A*). This dovetails with literature on spatial-navigation (*Doeller et al., 2008*; *Chersi and Burgess, 2015*; *Goodroe et al., 2018*; *Gahnstrom and Spiers, 2020*; *Geerts et al., 2020*; *Wiener et al., 2016*), showing that the striatum supports the reinforcement-dependent encoding of locations relative to landmarks, whereas the hippocampus may help to encode the structure of the environment in a generalizable and map-like format. This matches the functional differences observed here in the time domain, where caudate activity reflects the encoding of individual details of our task such as the TTC intervals (*Figure 3A*, *Figure 3—figure supplement 1A*, *Figure 3—figure supplement 1A*, *Figure 3—figure supplement 1B*), while the hippocampus generalizes across TTCs to encode the overall task structure (*Figure 3A*, *Figure 3B*, *Figure 3—figure supplement 1A*).

## Conclusion

In sum, we combined fMRI with time-to-contact estimations to show that the hippocampus supports the formation of task-specific yet flexible and generalizable sensorimotor representations in real time. Hippocampal activity reflected trial-wise behavioral feedback and the behavioral improvements across trials, suggesting that it supports sensorimotor updating even on short time scales. The observed updating signals were independent from the tested intervals, and they explained the regression-to-the-mean biases observed on a behavioral level. This is in line with the notion that the hippocampus encodes temporal context in a behavior-dependent manner, and that it supports finding an optimal trade off between specificity and regularization along with other regions. We show that it does so even in a fast-paced timing task typically considered to be hippocampal-independent. Our results show that the hippocampus supports rapid and feedback-dependent updating of sensorimotor representations, suggesting that it is a central component of a brain-wide network balancing task specificity vs. regularization for flexible behavior in humans.

# Materials and methods

## Participants

We recruited 39 healthy volunteers with normal to corrected-to-normal vision for this study (16 females, 19–35 years old). Five participants were excluded: one participant did not comply with the task instructions; one was excluded due to a failure of the eye-tracker calibration; three participants were excluded due to technical issues during scanning. A total of 34 participants entered the analysis. The sample size was chosen to accord with previous publications using similar procedures (*Nau et al., 2018a*; *Montchal et al., 2019*; *Schuck and Niv, 2019*). The study was approved by the regional committee for medical and health research ethics (project number 2017/969) in Norway and participants gave written consent prior to scanning in accordance with the declaration of Helsinki (*World Medical Association, 2013*).

## Task

Participants performed two tasks simultaneously: a smooth pursuit visual-tracking task and a time-to-contact estimation task. The visual tracking task entailed fixation at a fixation disc that moved on predefined linear trajectories with one of four speeds: 4.17°/s, 5.81°/s, 7.45°/s, and 9.09°/s. Upon reaching the end of such a linear trajectory, the dot stopped moving until the second task was completed. This second task was a time-to-collision (TTC) estimation task in which participants indicated when the fixation target would have hit a circular boundary if it had continued moving. This boundary was a yellow circular line surrounding the target trajectory with 10° radius. Participants gave their response by pressing a button at the anticipated moment of collision. They performed this task while still keeping fixation, and the individual linear trajectories were all of the same length (10° visual angle), leading to four target TTC durations of 1.2 s, 0.88 s, 0.67 s, and 0.55 s tested in a counterbalanced fashion across trials. After the button press, participants received feedback for 1 s informing them about the accuracy of their response. When participants *overestimated* the TTC, half of the fixation disc closest to the boundary changed color (orange or red) as a function of response accuracy (medium or low, respectively). When participants *underestimated* the TTC, half of the fixation

**Table 1.** Target TTCs' response windows for each feedback level.

| Target TTC = 0.55 s | |
| --- | --- |
| Accuracy | Response window (s) |
| High | 0.47–0.63 |
| Medium | 0.38–0.47 \| 0.63–0.71 |
| Low | <0.38 \| >0.71 |

| Target TTC = 0.67 s | |
| --- | --- |
| Accuracy | Response window (s) |
| High | 0.57–0.77 |
| Medium | 0.47–0.57 \| 0.77–0.87 |
| Low | <0.47 \| >0.87 |

| Target TTC = 0.86 s | |
| --- | --- |
| Accuracy | Response window (s) |
| High | 0.73–0.99 |
| Medium | 0.60–0.73 \| 0.99–1.12 |
| Low | <0.60 \| >1.12 |

| Target TTC = 1.2 s | |
| --- | --- |
| Accuracy | Response window (s) |
| High | 1.02–1.38 |
| Medium | 0.84–1.02 \| 1.38–1.56 |
| Low | <0.84 \| >1.56 |

disc further away from the boundary changed color. When participants were accurate, two opposing quadrants of the fixation disc would turn green. This allowed us to present feedback at fixation while keeping the number of informative pixels matched across feedback levels. To calibrate performance feedback across different TTC durations, the precise response window widths of each feedback level scaled with the speed of the fixation target (*Table 1*). The following formula was used to scale the response window width: $d \pm ((k * d)/2)$ where $d$ is the target TTC and $k$ is a constant proportional to 0.3 and 0.6 for high and medium accuracy, respectively. This ensured that participants received approximately the same feedback for tested TTCs despite the known differences in absolute performance between target TTCs due to inherent scalar variability (*Gibbon, 1977*). When no response was given, participants received low-accuracy feedback (two opposing quadrants of the fixation dot turned red) after a 4 s timeout. After the feedback, the disc remained in its last position for a variable inter-trial interval (ITI) sampled randomly from a uniform distribution between 0.5 s and 1.5 s. Following the end of the ITI, the dot continued moving in a different direction. In the course of 768 trials, each target TTC was sampled 192 times. We sampled eye-movement directions with 15° resolution, leading to an overall trajectory that was star-shaped, similar to earlier reports (*Nau et al., 2018a*). The full trajectory was never explicitly shown to the participants.

## Behavioral analysis

Participants indicated the estimated TTC in each trial via button press. In line with previous work (*Jazayeri and Shadlen, 2010*), participants tended to overestimate shorter durations and underestimate longer durations (*Figure 1B*). In order to quantify this behavioral effect we extracted the slope value of a linear regression line fit between estimated and target TTCs separately for each participant. A slope of 1 would indicate that participants performed perfectly accurately for all intervals. A slope of 0 would indicate that participants always gave the same response independent of the tested interval, fully regressing to the mean of the sampled intervals. Two separate one-tailed one-sample *t* tests (against 1 or 0) were performed to corroborate that participants' slope values regressed towards the mean of the sampled TTCs (*Figure 1—figure supplement 1D*). A Spearman's rank-order correlation tested if slope values correlated with the percent of high accuracy trials (*Figure 1—figure supplement 1E*), to further demonstrate that participants relied to different degrees on both, the target TTCs and the mean of the sampled TTCs, in order to achieve an optimal performance tradeoff. As the TTC task progressed, it would be expected that participants adjusted their TTC estimates in order to find the best tradeoff. Thus, we tested if the slope converged over time towards the value of 0.5 (the slope value between veridical performance and the mean of the sampled TTCs) by using a linear mixed-effects model with task segment as a predictor, the absolute difference between the slope and the value of 0.5 as the dependent variable and participants as the error term (*Figure 1—figure supplement 1F*). We also corroborated this effect by measuring the dispersion of slope values between participants across task segments using a linear regression model with task segment as a predictor and the standard deviation of slope values across participants as the dependent variable (*Figure 1—figure supplement 1G*). As a measure of behavioral performance, we computed two variables for each target-TTC level: sensorimotor timing accuracy, defined as the absolute difference in estimated and true TTC, and sensorimotor timing precision, defined as coefficient of variation (standard deviation of estimated TTCs divided by the average estimated TTC). To study the interaction between these two variables for each target TTC over time, we first normalized accuracy by the average estimated TTC in order to make both variables comparable. We then used a linear mixed-effects model with precision as the dependent variable, task segment and normalized accuracy as predictors and target TTC as the error term. In addition, we tested whether accuracy and precision increased over the course of the experiment using a linear mixed effects model with task segment as predictor and participants as the error term. Participants received feedback after each trial corresponding to the absolute TTC error of that trial. On average, 46.9% ($\sigma = 9.1$) of trials were of *high accuracy*, 31.2% ($\sigma = 3.9$) were of *medium* accuracy and 21.1% ($\sigma = 9.8$) were of *low* accuracy (*Figure 1C*). Moreover, we found that this feedback distribution was indeed similar across target-TTC levels as planned (*Figure 1—figure supplement 1B*), as well as across TTC over- and underestimation trials (*Figure 1—figure supplement 1C*). To control that there was no systematic and predictable relationship between subsequent trials on a behavioral level, we estimated the n-1 Pearson autocorrelation between feedback values received on each trial and then performed a two-tailed

one-sample t-test on group level against zero using the extracted correlation coefficients from each participant (*Figure 1—figure supplement 1A*).

## Imaging data acquisition and preprocessing

Imaging data were acquired on a Siemens 3T MAGNETOM Skyra located at the St. Olavs Hospital in Trondheim, Norway. A T1-weighted structural scan was acquired with 1 mm isotropic voxel size. Following EPI-parameters were used: voxel size = 2 mm isotropic, TR = 1020ms, TE = 34.6ms, flip angle = 55°, multiband factor = 6. Participants performed a total of four scanning runs of 16–18 min each including a short break in the middle of each run. Functional images were corrected for head motion and co-registered to each individual's structural scan using SPM12 (https://www.fil.ion.ucl.ac.uk/spm/). We used the FSL topup function to correct field distortions based on one image acquired with inverted phase-encoding direction (https://fsl.fmrib.ox.ac.uk/fsl/fslwiki/topup). Functional images were then spatially normalized to the Montreal Neurological Institute (MNI) brain template and smoothed with a Gaussian kernel with full-width-at-half-maximum of 4 mm for regions-of-interest analysis or with 8 mm for whole-brain analysis. Time series were high-pass filtered with a 128 s cut-off period. The results of all voxel-wise analyses were overlaid on a structural T1-template (colin27) of SPM12 for visualization.

## Regions of interest definition and analysis

Regions-of-interest masks for different brain areas were generated for each individual participant based on the automatic parcellation derived from FreeSurfer's structural reconstruction (https://surfer.nmr.mgh.harvard.edu/). The ROIs used in the present study include the Hippocampus as main area of interest (*Figure 2—figure supplement 1A*) as well as the Caudate Nucleus, Nucleus Accumbens, Thalamus, Putamen, Amygdala, and Globus Pallidum (*Figure 2—figure supplement 1B*). The hippocampal ROI was manually segmented following previous reports into its anterior and posterior sections based on the location of the uncal apex in the coronal plane as a bisection point (*Poppenk et al., 2013*). All individual ROIs were then spatially normalized to the MNI brain template space and re-sliced to the functional imaging resolution using SPM12. All ROI analyses were conducted using 4 mm spatial smoothing.

All ROI analyses described in the following were conducted using the following procedure. We extracted beta estimates estimated for the respective regressors of interest for all voxels within a region in both hemispheres, averaged them across voxels within that region and hemispheres and performed one-sample t-tests on group level against zero as implemented in the software R (https://www.R-project.org).

## Brain activity as a function of feedback on the previous trial

To examine how feedback modulates activity in the subsequent trial, we used a mass-univariate general linear model (GLM) analysis to model the activity of each voxel and trial as a function of feedback received in the previous trial. The GLM included three boxcar regressors modeling the trial period for each feedback level, plus one boxcar regressor for ITIs, one for button presses and one for periods of rest (inter-session interval, ISI), which were all convolved with the canonical hemodynamic response function of SPM12. The start of the trial was considered as the trial onsets for modeling (i.e. the time when the visual-tracking target started moving). The trial end was the offset of the feedback phase (i.e. the moment in which the feedback disappeared from the screen). The ITI was the time between the offset of the feedback-phase and the subsequent trial onset. In addition, the model included the six realignment parameters obtained during pre-processing as well as a constant term modeling the mean of the time series. On the group level, we then contrasted the weights obtained for the low-accuracy vs. high-accuracy feedback regressors and tested for differences using t-tests implemented in SPM12 (*Figure 2A*).

Additionally, we again conducted ROI analyses for the anterior and posterior sections of the hippocampus (*Figure 2—figure supplement 1A*) following the same procedure as described earlier (section "Regions of interest definition and analysis"). Here, we tested beta estimates obtained in the first-level analysis for the feedback-in-previous-trial regressor of interest (*Figure 2B*).

ITIs and ISIs were modeled to reduce task-unrelated noise, but to ensure that this did not lead to over-specification of the above-described GLM, we repeated the full analysis without modeling the

two. All other regressors including the main feedback regressors of interest remained unchanged, and we repeated both the voxel-wise and ROI-wise statistical tests as described above (*Figure 2—figure supplement 3B*).

Moreover, instead of modeling the three feedback levels with three independent regressors, we repeated the analysis modeling the three feedback levels as one parametric regressor with three levels. In addition, one boxcar regressor was added to model all trial periods independent from feedback level. All other regressors remained unchanged, and the model included the regressors for ITIs and ISIs. We then conducted t-tests implemented in SPM12 using the beta estimates obtained for the parametric feedback regressor (*Figure 2—figure supplement 3C*). Compared to the initial analyses presented above, this has the advantage that medium-accuracy feedback trials are considered for the statistics as well.

## Hippocampal functional connectivity as a function of previous-trial feedback

We conducted a psychophysiological interactions (PPI) analysis to examine whether hippocampal functional connectivity with the rest of the brain depended on the participant's performance on the previous trial. To do so, we centered a sphere onto the group-level peak effects within the HPC using main-effect GLM described in the previous section. The sphere was 4 mm in radius and was centered on the following MNI coordinates: x=-32, y=-14, z=-14. The GLM included a PPI regressor, a nuisance regressor accounting for the main effect of past-trial performance, and a nuisance regressor explaining variance due to inherent physiological signal correlations between the HPC and the rest of the brain. The PPI regressor was an interaction term containing the element-by-element product of the task time course (effects due to past-trial performance) and the HPC spherical seed ROI time course. The PPI model was built using the same model that revealed the main effects used to define the HPC sphere. The estimated beta weight corresponding to the interaction term was then tested against zero on the group-level using a t-test implemented in SPM12 (*Figure 2C*). The contrast reflects the difference between low vs. high-accuracy feedback. This revealed brain areas whose activity was co-varying with the hippocampus seed ROI as a function of past-trial performance (n-1).

## Brain activity as a function of current-trial performance and feedback

In two independent GLMs, we analyzed the time courses of all voxels in the brain as a function of behavioral performance (i.e. TTC error) in each trial, and as a function of feedback received at the end of each trial. The models included one mean-centered parametric regressor per run, modeling either the TTC error or the three feedback levels in each trial, respectively. Note that the feedback itself was a function of TTC error in each trial (high accuracy = 0, medium accuracy = 0.5 and low accuracy = 1). In addition, we added three nuisance regressors per run modeling ITIs, button presses, and periods of rest. These regressors were convolved with the canonical hemodynamic response function of SPM12. Moreover, the realignment parameters and a constant term were again added. We estimated weights for all regressors and conducted a t-test against zero using SPM12 for our feedback and performance regressors of interest on the group level (*Figure 2—figure supplement 2A*). Importantly, positive t-scores indicate a positive relationship between fMRI activity and TTC error and hence with poor behavioral performance. Conversely, negative t-scores indicate a negative relation between the two variables and hence better behavioral performance.

In addition to the voxel-wise whole-brain analyses described above, we conducted independent ROI analyses for the anterior and posterior sections of the hippocampus (*Figure 2—figure supplement 1A*). Here, we tested the beta estimates obtained in our first-level analysis for the feedback and performance regressors of interest (*Figure 2—figure supplement 2B*; two-tailed one-sample $t$ tests: anterior HPC, $t(33) = -5.92, p = 1.2x10^{-6}, p_{fwe} = 2.4x10^{-6}, d = -1.02, CI : [-1.45, -0.6]$; posterior HPC, $t(33) = -4.07, p = 2.7x10^{-4}, p_{fwe} = 5.4x10^{-4}, d = -0.7, CI : [-1.09, -0.32]$). See section 'Regions of interest definition and analysis' for more details.

## Brain activity as a function of improvements in behavioral performance across trials

We used a GLM to analyze activity changes associated with behavioral improvements across trials. One regressor modeled the main effect of the trial and two parametric regressors modeled the

following contrasts: Parametric regressor 1: trials in which behavioral performance improved *vs.* parametric regressor 2: trials in which behavioral performance did not improve or got worse relative to the previous trial. These regressors modeled the behavioral improvements either relative to the previous trial, and therefore independently of TTC (likely serving regularization), or relative to the previous trial in which the same target TTC was presented (likely serving specificity). These two regressors reflect the tests for target-TTC-independent and target-TTC-specific updating, respectively, and they were not orthogonalized to each other. Because we predicted to find stronger activity for improvements vs. no improvements in behavioral performance, we here performed one-tailed statistical tests, consistent with the direction of this hypothesis. Improvement in performance was defined as receiving feedback of higher valence than in the corresponding previous trial. The same nuisance regressors were added as in the other GLMs and all regressors except the realignment parameters and the constant term were convolved with the canonical hemodynamic response function of SPM12. On the group level, we tested the two parametric regressors of interest against zero using a t-test implemented in SPM12, effectively contrasting trials in which behavioral performance improved against trials in which behavioral performance did not improve or got worse relative to the respective previous trials (*Figure 3A*). All runs were modeled separately.

Moreover, we again conducted ROI analyses for the anterior and posterior sections of the hippocampus (*Figure 2—figure supplement 1A*) following the same procedure as described earlier (see section 'Regions of interest definition and analysis'). Here, we tested beta estimates obtained in the first-level analysis for the TTC-specific and TTC-independent updating regressors using one-tailed one-sample t-tests (*Figure 3B*). In addition, to test which specific subcortical regions were involved in these processes, we conducted post-hoc ROI analyses for subcortical regions after the whole-brain results were known (*Figure 3—figure supplement 1B*; one-tailed one-sample $t$ tests; TTC-specific: caudate: $t(33) = 5.95$, $p = 5.6x10^{-7}$, $p_{fwe} = 3.4x10^{-6}$, $d = 1.02$, $CI : [0.61, 1.45]$, nucleus accumbens: $t(33) = 4.41$, $p = 5.2x10^{-5}$, $p_{fwe} = 3.1x10^{-4}$, $d = 0.76$, $CI : [0.38, 1.15]$, globus pallidus: $t(33) = 7.05$, $2.3x10^{-8}$, $p_{fwe} = 1.4x10^{-7}$, $d = 1.21$, $CI : [0.77, 1.67]$, putamen: $t(33) = 8.07$, $p = 1.3x10^{-9}$, $p_{fwe} = 7.7x10^{-9}$, $d = 1.38$, $CI : [0.92, 1.88]$, amygdala: $t(33) = 1.78$, $p = 0.042$, $p_{fwe} = 0.255$, $d = 0.30$, $CI : [-0.04, 0.66]$, thalamus: $t(33) = 2.61$, $p = 0.007$, $p_{fwe} = 0.007$, $d = 0.45$, $CI : [0.09, 0.81]$; TTC-independent: caudate: $t(33) = -0.67$, $p = 0.746$, $p_{fwe} = 1$, $d = -0.11$, $CI : [-0.46, 0.23]$, nucleus accumbens: $t(33) = 1.82$, $p = 0.039$, $p_{fwe} = 0.235$, $d = 0.31$, $CI : [-0.04, 0.66]$, globus pallidus: $t(33) = 7.06$, $p = 2.2x10^{-8}$, $p_{fwe} = 1.3x10^{-7}$, $d = 1.21$, $CI : [0.77, 1.68]$, putamen: $t(33) = 6.21$, $p = 2.6x10^{-7}$, $p_{fwe} = 1.6x10^{-6}$, $d = 1.06$, $CI : [0.65, 1.50]$, amygdala: $t(33) = 4.25$, $p = 8.3x10^{-5}$, $p_{fwe} = 4.9x10^{-4}$, $d = 0.73$, $CI : [0.35, 1.12]$, thalamus: $t(33) = 4.05$, $p = 1.5x10^{-4}$, $p_{fwe} = 8.9x10^{-4}$, $d = 0.69$, $CI : [0.32, 1.08]$). The subcortical ROIs (*Figure 2—figure supplement 1B*) were based on the FreeSurfer parcellation as described in the section 'Regions of interest definition and analysis'.

## Hippocampal functional connectivity as a function of TTC-independent updating

To examine which brain regions whose activity co-fluctuated with the one of the hippocampus during TTC-independent updating, we again conducted a PPI analysis similar to the one described earlier. A spherical seed ROI with a radius of 4 mm was centered around the hippocampal group-level peak effect (x=-30, y=-24, z=-18) observed for the TTC-independent updating regressor described above. The GLM included a PPI regressor and two nuisance regressors accounting for task-related effects from our contrast of interest (Behavioral improvements vs. no behavioral improvements) as well as physiological correlations that could arise due to anatomical connections to the hippocampal seed region or shared subcortical input. On the group-level, we then tested the weights estimated for our PPI regressor of interest against zero using a t-test implemented in SPM12. This revealed areas whose activity co-fluctuated with the one of the hippocampus as a function TTC-independent updating (*Figure 3—figure supplement 2A*).

Moreover, we conducted independent ROI analyses for subcortical regions as described in the section 'Regions of interest definition and analysis'. Here, we tested the beta estimates obtained for the hippocampal seed-based PPI regressor of interest (*Figure 3—figure supplement 2B*; one-tailed one-sample $t$ tests: caudate: $t(33) = 1.06$, $p = 0.149$, $p_{fwe} = 0.894$, $d = 0.18$, $CI : [-0.16, 0.53]$, putamen: $t(33) = 2.79$, $p = 0.004$, $p_{fwe} = 0.026$, $d = 0.48$, $CI : [0.12, 0.84]$ globus pallidus: $t(33) = 2.52$, $p = 0.008$, $p_{fwe} = 0.050$, $d = 0.43$, $CI : [0.08, 0.79]$, amygdala: $t(33) = 2.60$, $p = 0.007$, $p_{fwe} = 0.041$, $d = 0.45$, $CI : [0.09, 0.81]$, nucleus

accumbens:$t(33) = -1.14, p = 0.869, p_{fwe} = 1, d = -0.20, CI : [-0.54, 0.15]$,thalamus:$t(33) = 2.71, p = 0.005, p_{fwe} = 0.032, d = 0.46, CI : [0.11, 0.83]$).

## Brain activity as a function of behavioral performance and as a function of the behavioral regression effect

To examine the neural underpinnings governing specificity and regularization in timing behavior in detail, we analyzed the trial-wise activity of each voxel as a function of performance in the TTC task (i.e. the absolute difference between estimated and true TTC in each trial) and as a function of the regression effect in behavior (i.e. the absolute difference between the estimated TTC and the mean of the sampled intervals, which was 0.82 s). To avoid effects of potential co-linearity between these regressors, we estimated model weights using two independent GLMs, which modeled the time course of each trial with either one of the two regressors. In addition, we again accounted for nuisance variance as described before, and all regressors except the realignment parameters and the constant term were convolved with the canonical HRF of SPM12. After fitting the model, we used the weights estimated for the two regressors to perform voxel-wise F-tests using SPM12, revealing activity that was correlated with these two regressors independent of the sign of the correlation (*Figure 4A*). In addition, we again performed ROI analyses using two-tailed one-sample t-tests for the anterior and posterior hippocampus (*Figure 2—figure supplement 1A*, *Figure 4B*).

## Eye tracking: Fixation quality does not affect the interpretation of our results

We used an MR-compatible infrared eye tracker with long-range optics (Eyelink 1000) to monitor gaze position at a rate of 500 hz during the experiment. After blink removal, the eye tracking data was linearly detrended, median centered, downsampled to the screen refresh rate of 120 hz and smoothed with a running-average kernel of 100ms. Wilcoxon signed-rank tests for paired samples were used in order to test for potential biases in fixation error across speeds (*Figure 1—figure supplement 2A*) or across feedback levels (*Figure 1—figure supplement 2B*). Moreover, we tested if differences in fixation error could either explain individual differences in the regression effect, or individual differences in absolute TTC error in behavior using Spearman's rank-order correlations (*Figure 1—figure supplement 2C*).

## Acknowledgements

We thank Raymundo Machado de Azevedo Neto for helpful comments on an earlier version of this manuscript. CFD's research is supported by the Max Planck Society, the Kavli Foundation, the Jebsen foundation, the Centre of Excellence scheme of the Research Council of Norway – Centre for Neural Computation (223262 /F50), The Egil and Pauline Braathen and Fred Kavli Centre for Cortical Micro-circuits and the National Infrastructure scheme of the Research Council of Norway – NORBRAIN (197467 /F50). MN's research is supported by a Feodor-Lynen Research Fellowship of the Alexander von Humboldt Foundation. RK's research is supported by a CIDEGENT grant (CIDEGENT/2021/027) from the Valencian Community's program for the support of talented researchers and the Ministerio de Ciencia, Innovación y Universidades, which is part of the Agencia Estatal de Investigación (AEI), through the project PID2021-12233NA-100.

## Additional information

### Competing interests
Virginie van Wassenhove: Reviewing editor, *eLife*. The other authors declare that no competing interests exist.

## Funding

| Funder | Grant reference number | Author |
| --- | --- | --- |
| European Research Council | ERC-CoG GEOCOG 724836 | Christian F Doeller |
| Max Planck Society | | Christian F Doeller |
| Kavli Foundation | | Christian F Doeller |
| Kristian Gerhard Jebsen Foundation | | Christian F Doeller |
| Norges Forskningsråd | 223262/F50 | Christian F Doeller |
| Egil and Pauline Braathen and Fred Kavli Centre for Cortical Microcircuits | | Christian F Doeller |
| Norges Forskningsråd | NORBRAIN 197467/F50 | Christian F Doeller |
| Alexander von Humboldt Foundation | Feodor-Lynen Research Fellowship | Matthias Nau |
| Generalitat Valenciana | CIDEGENT/2021/027 | Raphael Kaplan |
| Ministerio de Ciencia, Innovación y Universidades | PID2021-122338NA-I00 | Raphael Kaplan |
| Commissariat à l'Énergie Atomique et aux Énergies Alternatives | | Virginie van Wassenhove |
| Institut National de la Santé et de la Recherche Médicale | | Virginie van Wassenhove |

The funders had no role in study design, data collection and interpretation, or the decision to submit the work for publication.

## Author contributions

Ignacio Polti, Conceptualization, Data curation, Software, Formal analysis, Investigation, Visualization, Methodology, Writing – original draft, Project administration, Writing – review and editing; Matthias Nau, Conceptualization, Data curation, Supervision, Visualization, Methodology, Writing – original draft, Project administration, Writing – review and editing; Raphael Kaplan, Supervision, Project administration, Writing – review and editing; Virginie van Wassenhove, Supervision, Writing – review and editing; Christian F Doeller, Conceptualization, Supervision, Funding acquisition, Project administration, Writing – review and editing

## Author ORCIDs

Ignacio Polti http://orcid.org/0000-0002-6631-4315
Matthias Nau http://orcid.org/0000-0003-0956-7815
Raphael Kaplan http://orcid.org/0000-0002-5023-1566
Virginie van Wassenhove http://orcid.org/0000-0002-2569-5502
Christian F Doeller http://orcid.org/0000-0003-4120-4600

## Ethics

The study was approved by the regional committee for medical and health research ethics (project number 2017/969) in Norway and participants gave written consent prior to scanning in accordance with the declaration of Helsinki (World Medical Association, 2013).

## Decision letter and Author response

Decision letter https://doi.org/10.7554/eLife.79027.sa1
Author response https://doi.org/10.7554/eLife.79027.sa2

## Additional files

### Supplementary files
• MDAR checklist

### Data availability

Source data and analysis code are available at the following Open Science Framework repository: https://osf.io/cs8d6/. Pre-processed eye-tracker data can be found here: https://osf.io/mrhk9. Raw fMRI data is available at the following G-Node Infrastructure repository: https://gin.g-node.org/ipolti/TTC_HPCF.git.

The following datasets were generated:

| Author(s) | Year | Dataset title | Dataset URL | Database and Identifier |
| --- | --- | --- | --- | --- |
| Polti I, Nau M, Kaplan R, Wassenhove van, Doeller CF | 2022 | Time-To-Contact | https://gin.g-node.org/ipolti/TTC_HPCF.git | G-Node Infrastructure, 10.12751/g-node.pwn4qz |
| Frey M, Nau M, Doeller CF | 2021 | DeepMReye | https://osf.io/mrhk9/ | Open Science Framework, 10.17605/OSF.IO/MRHK9 |
| Polti I, Nau M | 2022 | Rapid encoding of task regularities in the human hippocampus guides sensorimotor timing | https://osf.io/cs8d6/ | Open Science Framework, cs8d6 |

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
