## [Editor Report]

This important work brings ideas about hippocampal learning and involvement in temporal processing to a sensorimotor timing task, "time-to-contact estimation", that is not typically considered to be hippocampus-dependent. The study found that activity in the hippocampus measured with fMRI was related to feedback received about the accuracy of timing estimation and to performance improvement across trials in a manner not tied to the specific time interval tested. The evidence presented for the nature of the involvement of the hippocampus in this task is compelling.

---

## [Decision Letter]

**Decision letter after peer review:**

Thank you for submitting your article "Rapid encoding of task regularities in the human hippocampus guides sensorimotor timing" for consideration by *eLife*. Your article has been reviewed by 3 peer reviewers, one of whom is a member of our Board of Reviewing Editors, and the evaluation has been overseen by Tamar Makin as the Senior Editor. The reviewers have opted to remain anonymous.

Essential revisions:

1) Adjust framing/speculation as recommended by Reviewers 1 and 3.

2) Carry out the analyses suggested by Reviewer 2.

3) Significantly clarify the methods as suggested by Reviewer 3.

4) All three reviewers were confused about the approach towards modeling high, medium, and low feedback. Both clarification and new analyses are likely needed to address this.

*Reviewer #1 (Recommendations for the authors):*

The accuracy measure is collapsed across overestimation and underestimation. It might be useful to demonstrate that the results are the same for under and overestimation.

[Editors' note: further revisions were suggested prior to acceptance, as described below.]

Thank you for resubmitting your work entitled "Rapid encoding of task regularities in the human hippocampus guides sensorimotor timing" for further consideration by *eLife*. Your revised article has been evaluated by Tamar Makin (Senior Editor) and a Reviewing Editor.

The manuscript has been much improved. Two of the reviewers were satisfied that their concerns were addressed, but one reviewer has remaining concerns that need to be addressed:

I would like to thank the authors for the thorough revision of the manuscript, for running the additional control analyses that were suggested and for considering my comments so carefully.

While the analytical pipeline has been clarified, there are still some inconsistencies in the text/figure and in the response document that warrant further clarification. Specifically, the authors confirmed that the results presented in Figure 2 correspond to a GLM model with 3 regressors without parametric modulators and that the contrast presented in panel A corresponds to a HIGH > LOW contrast. It is now stated that "We plot the β estimates obtained for the contrast between high vs. low feedback", thus a subtraction between β high and β low. Therefore – without any parametric modulator in the model – a negative β value (as observed in the hippocampus) corresponds to greater activity in low vs. high feedback trials (LOW > HIGH). In contrast, it is stated in the legend of Figure 2 that "Negative values indicate that smaller errors, and higher-accuracy feedback, led to stronger activity" which might still refer to a parametric approach as used later in the manuscript. This point is critical to address as the results presented in Figure 2 – if based on a HIGH > LOW contrast as mentioned in the response document and the revised manuscript – are in line with prediction error processes (greater activity for larger errors). It would be helpful to depict the β values for each regressor included in the contrast (i.e., high and low) to better evaluate the direction of the effect instead of plotting the difference between the two if it's what panel B is indeed showing.

The same comment applies to the PPI results. The authors clarify that the PPI results correspond to the contrast LOW vs. HIGH. Color coding is confusing in Figure 2 as panel A displays HIGH > LOW and panel C depicts HOW > HIGH using same color coding (yellow for positive effect).

I would also like to thank the authors for running the additional analyses with performance (as opposed to feedback) as a parametric modulator. The results are indeed very similar but in the revised Figure 2—figure supplement 2 panel C, the sagittal view does not highlight the hippocampal cluster anymore.

The GLM description is still unclear. Based on the new information provided by the authors, the three box car regressors are ITI, button press and ISI. I assume that the task was also modelled with a box car as the authors clarified the onset and offset of the task in the methods. Therefore, it appears that the GLM included 4 box car regressors per feedback level. Is that correct? Please clarify.

---

## [Author Response]

We extensively revised the description of this main analysis to ensure that the motivation behind it and the interpretations of its results are clear. This includes changes in the method section and in the main text.In addition, and as mentioned before, two new fMRI analyses show that the key result (i.e. hippocampal activity reflects behavioral performance-dependent feedback) is robust even when (1) feedback is modeled with one parametric regressor instead of three independent regressors, and (2) when task performance is modeled instead of feedback. All conclusions remain unchanged.Reviewer #1 (Recommendations for the authors):The accuracy measure is collapsed across overestimation and underestimation. It might be useful to demonstrate that the results are the same for under and overestimation.

In response to the reviewer’s comment, we added a new Figure 1—figure supplement 1C, showing that there were no systematic differences in received feedback between overestimation and underestimation.

The following addition was made to the Methods section:

Page 14: “Moreover, we found that this feedback distribution was indeed similar across target-TTC levels as planned (Figure 1—figure supplement 1B), as well as across TTC over- and underestimation trials (Figure 1—figure supplement 1C).”

[Editors' note: further revisions were suggested prior to acceptance, as described below.]

The manuscript has been much improved. Two of the reviewers were satisfied that their concerns were addressed, but one reviewer has remaining concerns that need to be addressed:I would like to thank the authors for the thorough revision of the manuscript, for running the additional control analyses that were suggested and for considering my comments so carefully.While the analytical pipeline has been clarified, there are still some inconsistencies in the text/figure and in the response document that warrant further clarification. Specifically, the authors confirmed that the results presented in Figure 2 correspond to a GLM model with 3 regressors without parametric modulators and that the contrast presented in panel A corresponds to a HIGH > LOW contrast. It is now stated that "We plot the β estimates obtained for the contrast between high vs. low feedback", thus a subtraction between β high and β low. Therefore – without any parametric modulator in the model – a negative β value (as observed in the hippocampus) corresponds to greater activity in low vs. high feedback trials (LOW > HIGH). In contrast, it is stated in the legend of Figure 2 that "Negative values indicate that smaller errors, and higher-accuracy feedback, led to stronger activity" which might still refer to a parametric approach as used later in the manuscript. This point is critical to address as the results presented in Figure 2 – if based on a HIGH > LOW contrast as mentioned in the response document and the revised manuscript – are in line with prediction error processes (greater activity for larger errors). It would be helpful to depict the β values for each regressor included in the contrast (i.e., high and low) to better evaluate the direction of the effect instead of plotting the difference between the two if it's what panel B is indeed showing.

We are extremely grateful to the reviewer for highlighting this relevant inconsistency, and we apologize for missing to correct it in the first round of revisions. The reviewer is right that the contrast description in the figure caption was still wrong. We corrected it in both Figure 2 and Figure 2 —figure supplement 3, which now correctly refer to the contrast “low-accuracy vs. high-accuracy feedback”. We also corrected a mention of the contrast in the Results section (page 5). All other mentions of this contrast were correct and our interpretations remain unchanged.

We further adapted one sentence in the methods section (page 15) to avoid potential confusion. Instead of referring to “TTC error” here, we now refer to “accuracy”, making this contrast description more consistent with other mentions in the text. The new sentence reads as follows.

“On the group level, we then contrasted the weights obtained for the low-accuracy vs. high-accuracy feedback regressors […]”

The same comment applies to the PPI results. The authors clarify that the PPI results correspond to the contrast LOW vs. HIGH. Color coding is confusing in Figure 2 as panel A displays HIGH > LOW and panel C depicts HOW > HIGH using same color coding (yellow for positive effect).

We would like to thank the referee. There was a mistake in the description of Figure 2A. The description of the contrast in Figure 2C was correct. We plot the contrast between low-accuracy vs. high-accuracy feedback in both Figure 2A and Figure 2C (the former of which has been corrected).

I would also like to thank the authors for running the additional analyses with performance (as opposed to feedback) as a parametric modulator. The results are indeed very similar but in the revised Figure 2—figure supplement 2 panel C, the sagittal view does not highlight the hippocampal cluster anymore.

That is correct. It is consistent with our proposal that it is the feedback that is reflected in hippocampal activity, not the task performance per se, possibly reflecting sensorimotor updating that should occur when feedback is received. Performance and feedback were tightly correlated (the former defined the latter), which may explain why the whole-brain voxel-wise maps and the hippocampal ROI results look extremely similar nevertheless. Note that also the hippocampal cluster is visible if the data was plotted at uncorrected thresholds (Author response image 1).

**Author response image 1. sa2fig1:** Left: Effect of task performance modeled with one parametric regressor (Figure 2—figure supplement 2). Right: Same data as on the left but plotted at a more liberal, uncorrected threshold. Only negative t-scores shown on the right.

The GLM description is still unclear. Based on the new information provided by the authors, the three box car regressors are ITI, button press and ISI. I assume that the task was also modelled with a box car as the authors clarified the onset and offset of the task in the methods. Therefore, it appears that the GLM included 4 box car regressors per feedback level. Is that correct? Please clarify.

The GLM modeling the 3 feedback levels with independent regressors (Figure 2) included 3 additional nuisance boxcar regressors (ISI, button presses, ITIs). This is now described on Page 15 as follows:

“The GLM included three boxcar regressors modeling the trial period for each feedback level, plus one boxcar regressor for ITIs, one for button presses and one for periods of rest (inter-session in-terval, ISI) […]”

The GLM modeling feedback with one parametric regressor (Figure 2 —figure supplement 3) included one boxcar regressor for all trial periods as well as a feedback parametric modulator, and it additionally included the same 3 nuisance boxcar regressors mentioned above. We clarified this in the methods section on Page 16, which now reads as follows.

“Moreover, instead of modeling the three feedback levels with three independent regressors, we repeated the analysis modeling the three feedback levels as one parametric regressor with three levels. In addition, one boxcar regressor was added to model all trial periods independent from feedback level."